# Automatic Variable Selection Algorithms in Prognostic Factor Research in Neck Pain

**DOI:** 10.3390/jcm12196232

**Published:** 2023-09-27

**Authors:** Bernard X. W. Liew, Francisco M. Kovacs, David Rügamer, Ana Royuela

**Affiliations:** 1School of Sport, Rehabilitation and Exercise Sciences, University of Essex, Colchester CO4 3SQ, Essex, UK; 2Unidad de la Espalda Kovacs, HLA-Moncloa University Hospital, 28008 Madrid, Spain; fmkovacs@kovacs.org; 3Department of Statistics, Ludwig-Maximilians-Universität München, 80539 Munich, Germany; david.ruegamer@stat.uni-muenchen.de; 4Biostatistics Unit, Hospital Puerta de Hierro, Instituto Investigación Sanitaria Puerta de Hierro-Segovia de Arana, Consorcio de Investigación Biomédica en Red de Epidemiología y Salud Pública, Red Española de Investigadores en Dolencias de la Espalda, 28222 Madrid, Spain; aroyuela@idiphim.org

**Keywords:** neck pain, statistics, prognosis, machine learning, variable selection

## Abstract

This study aims to compare the variable selection strategies of different machine learning (ML) and statistical algorithms in the prognosis of neck pain (NP) recovery. A total of 3001 participants with NP were included. Three dichotomous outcomes of an improvement in NP, arm pain (AP), and disability at 3 months follow-up were used. Twenty-five variables (twenty-eight parameters) were included as predictors. There were more parameters than variables, as some categorical variables had >2 levels. Eight modelling techniques were compared: stepwise regression based on unadjusted *p* values (stepP), on adjusted *p* values (stepPAdj), on Akaike information criterion (stepAIC), best subset regression (BestSubset) least absolute shrinkage and selection operator [LASSO], Minimax concave penalty (MCP), model-based boosting (mboost), and multivariate adaptive regression splines (MuARS). The algorithm that selected the fewest predictors was stepPAdj (number of predictors, *p* = 4 to 8). MuARS was the algorithm with the second fewest predictors selected (*p* = 9 to 14). The predictor selected by all algorithms with the largest coefficient magnitude was “having undergone a neuroreflexotherapy intervention” for NP (β = from 1.987 to 2.296) and AP (β = from 2.639 to 3.554), and “Imaging findings: spinal stenosis” (β = from −1.331 to −1.763) for disability. Stepwise regression based on adjusted *p*-values resulted in the sparsest models, which enhanced clinical interpretability. MuARS appears to provide the optimal balance between model sparsity whilst retaining high predictive performance across outcomes. Different algorithms produced similar performances but resulted in a different number of variables selected. Rather than relying on any single algorithm, confidence in the variable selection may be increased by using multiple algorithms.

## 1. Introduction

Neck pain (NP) is a very common musculoskeletal pain disorder [1] that not only results in considerable pain and suffering, but incurs a significant economic cost [2]. The management of NP is complex given the multifactorial nature of the disorder [3]. Prognostic factor research [4] is seen as key to disentangling the complexity of NP, by identifying predictors of poor outcomes for treatment [5]. Recent systematic reviews have identified several prognostic factors of poor outcome in NP, which include body mass index (BMI) [5], fear [6], NP intensity at inception [6], and symptom duration [7], to name a few.

Multivariable statistical models are commonly used in prognostic factor research [8,9]. To identify the most important variables as predictors, the most common statistical strategy is stepwise regression, where only variables where the statistical significance exceeds a threshold are retained as predictors [10,11,12,13]. It has long been recognised that the standard errors of the coefficient estimates are underestimated when standard statistical tests, which assume a single test of a pre-specified model, are applied sequentially like in a stepwise regression [14]. This could result in variables being more likely to be retained because of an artificially small *p* value. The potential that less important variables are included into the model could reduce prediction performance in the testing (out-of-sample) data [14].

Increasingly, machine learning (ML) is being employed for prognostic modelling [15,16]. A significant barrier to embedding ML models into mainstream clinical care is its “black-box” approach [17]. The lack of model interpretability means that a clinician cannot decide how the model reached its final prediction. In contrast to ML, statistical methods like logistic/linear regression are intrinsically interpretable, given that from the magnitude and sign of the coefficient estimates of the included predictors, the predicted outcome can be determined. However, there are interpretable ML algorithms that perform automatic variable selection during the model fitting process, such as model-based boosting (mboost) [18], the least absolute shrinkage and selection operator (LASSO) [19], and multivariate adaptive regression spline (MuARS) [20], to name a few. ML algorithms that perform intrinsic automatic variable selection are known as embedded strategies [21]. Filter-based strategies reflect preprocessing steps that use a criterion not involved in any ML algorithms, to preselect a subset of all candidate variables, to be used in ML [21]. An example of filter-based strategies includes removing highly collinear variables before ML modelling. In wrapper-based strategies, the variable selection is based on a specific ML algorithm, which follows a greedy search by evaluating all possible combinations of variables against the evaluation criterion [21]. An example of wrapper-based approaches includes stepwise selection using the Akaike information criterion (AIC).

We previously compared different “black-box” ML algorithms against traditional statistical methods [22]. No studies to date have compared the differences in variables selected and the magnitude and sign of their coefficient estimates between different ML algorithms against traditional stepwise regression for NP prognostic factor studies. Hence, the primary aim of this study is to compare how different ML and statistical algorithms differ in the number of variables selected, and the associated magnitude and sign of the estimated coefficients. Herein, we restricted the comparison to parametric ML algorithms with embedded variable selection capacity, as well as wrapper methods [23]. The secondary aim of this study is to compare how differences in the variables selected and their coefficient estimates between different ML and statistical algorithms influence the prediction performance of these algorithms. We first hypothesised that traditional stepwise regression using unadjusted *p* values would lead to the least sparse model. We also hypothesised that the prediction performance of traditional stepwise regression using unadjusted *p* values would be the poorest compared to the remaining ML algorithms assessed.

## 2. Materials and Methods

### 2.1. Design

This was a longitudinal observational study with repeated measurements at baseline and at 3 months follow-up. This study follows the transparent reporting of a multivariable prediction model for individual prognosis or diagnosis (TRIPOD) statement [24].

### 2.2. Setting

Forty-seven health care centres were invited by the Spanish Back Pain Research Network to participate in this study [8]. According to Spanish law (Ley de Investigación Biomédica 14/2007 de 3 de julio, ORDEN SAS/3470/2009, de 16 de diciembre-BOE núm. 310, de 25 diciembre [RCL 2009, 2577]-), no ethical approval was required due to the observational design of this study.

### 2.3. Participants

The recruitment window spanned the period from 2014 to 2017 [8]. The inclusion criteria were participants suffering from non-specific NP, with or without arm pain, seeking care for NP in a participating unit, and fluent in the Spanish language. The exclusion criteria were participants suffering from any central nervous system disorders, and where NP or arm pain were due to trauma or a specific systemic disease.

### 2.4. Sample Size

The sample size was established at 2934 subjects. There were no concerns about the sample size being too large, due to the observational nature of the study. To analyse the association of up to 40 parameters, the sample had to include at least 400 subjects who would not experience improvement, following the 1:10 (1 parameter per 10 events) rule of thumb [25].

### 2.5. Predictor and Outcome Variables

Data collected at baseline from participants included age, sex, duration of the current pain episode (days), the time elapsed since the first episode (years), and work status. At baseline and follow-up, participants were asked to report the intensity of their neck and arm pain and neck-related disability. For pain intensity measurements, 10 cm visual analog scales (VAS) were used (0  =  no pain and 10  =  worst imaginable pain). For disability, the Spanish version of the Neck Disability Index (NDI, 0  =  no disability and 100  =  worst possible disability) [26] was used (Table 1).

Data collected at baseline from clinicians included diagnostic procedures provided for the current episode (e.g., X-rays, computed tomography (CT) scans, magnetic resonance imaging (MRI)), radiological reports of the current or previous episodes (e.g., facet joint degeneration, spinal stenosis), clinical diagnosis (pain caused by disc herniation, spinal stenosis or “non-specific NP”), and treatments received by the participant (e.g., drugs—analgesics, NSAIDs; physiotherapy and rehabilitation; neuroreflexotherapy intervention; surgery) (Table 1).

Three binary outcomes were analysed in this study: NP, AP, and NDI improvements (yes/no), all at the 3rd month follow-up. Most of the improvements in people with spinal pain disorders occur within the first 3 months. Also, there is a substantial attrition of patients after 3 months’ follow-up [27,28]. Hence, the primary outcomes were collected on the 3rd month follow-up. An improvement was defined if the reductions in VAS or NDI scores between the baseline and follow-up assessments were greater than the minimal clinically important change (MCIC), i.e., a minimum value of 1.5 for VAS and 7 NDI points [26].

### 2.6. Preprocessing and Missing Data Handling

Figure 1 provides a schematic illustration of the workflow in this study. Twenty-five variables were included in the present study. The data (*n* = 3001) were split into a training set (80%, *n* = 2402) and testing set (20%, *n* = 599) for validation. Multiple imputation by chained equations method [29] was performed given that no systematic patterns of missing data were noted. Multiple imputations on the training set were performed. The imputed model was then used to impute missing data in the testing set. All five continuous predictors were scaled to have a mean of zero and a standard deviation (SD) of one. All 20 categorical variables were transformed into integers using one-hot encoding. Altogether, there were 28 parameters included as predictors in the model, without considering the intercept.

### 2.7. ML Algorithms

The codes used for the present study are included in the lead author’s public repository (https://github.com/bernard-liew/spanish_data_repo accessed on 18 September 2023). Eight algorithms were compared in the present study and their details can be found in the Appendix A: (1) stepwise logistic regression based on *p* values with no adjustment (stepP) [30]; (2) stepwise logistic regression based on *p* values with adjustment (stepPAdj) [31]; (3) stepwise logistic regression based on AIC (stepAIC) [32]; (4) best subset regression (BestSubset) [33]; (5) LASSO [19,24]; (6) Minimax concave penalty (MCP) regression; (7) model-based boosting (mboost) [18]; and (8) MuARS [20]. Both LASSO and mboost produce coefficients that are biased towards zero [18]. Hence, the predictors selected by LASSO and mboost were refitted with a simple logistic regression model to retrieve the unbiased coefficients. Stepwise regression methods were selected as they represent the most traditional methods used in spinal pain research for variable selection [34,35]. Regularised regression methods (e.g., LASSO, MCP, boosting) have been advocated as preferable techniques used for variable selection by TRIPOD [24]. MuARS was selected based on its optimal balance between model sparsity and prediction performance in prior research in a similar disease cohort [36]. BestSubset was used based on prior research on its superior predictive performance and quicker computational speed compared to traditional regularised methods [37].

### 2.8. Validation

The primary measure of model performance was the area under the curve (AUC) of the testing set [22]. AUC ranges from 0 to 1, with a value of 1 being when the model can perfectly classify all the improvements and no improvements correctly. The secondary measures of performance were classification accuracy, precision, sensitivity, specificity, and the F1 score, as described in a prior study [22]. We were also interested in exploring the sparsity of each modelling algorithm, and whether the number of selected coefficients, its coefficient magnitude, and sign were similar across algorithms.

## 3. Results

The descriptive characteristics of participants can be found in Table 1. Across the three outcomes, the algorithm that selected the fewest predictors was stepPAdj (number of predictors, *p* = 4 to 8), whilst the algorithm that selected the greatest number of predictors was LASSO for the outcomes of AP and disability (*p* = 21 to 28) and the best subset for NP (*p* = 28) (Table 2, Table 3 and Table 4, see Appendix A). MuARS was the algorithm with the second fewest predictors selected (*p* = 9 to 14) (Table 2, Table 3 and Table 4, Appendix A). For the outcomes of NP, AP, and disability, three, three, and six predictors were selected by all eight algorithms (Table 2, Table 3 and Table 4). Two variables that were not selected by either of the two *p*-value-based stepwise regressions were selected by the remaining six algorithms for the outcome of NP; eight variables followed the same trend for the outcome of AP; and four variables followed this trend for disability (Table 2, Table 3 and Table 4, Appendix A).

For the outcome of NP, the difference in predictive performance between the best and worst algorithms was small, with a difference of 0.01, 0.02, 0.04, 0.03, and 0.01 for accuracy, AUC, precision, sensitivity, and specificity, respectively (Figure 2A, see Appendix A). For the outcome of AP, the difference in predictive performance between the best and worst algorithms was 0.01, 0.02, 0.03, 0.06, and 0.03 for accuracy, AUC, precision, sensitivity, and specificity, respectively (Figure 2B, Appendix A). For disability, the difference in predictive performance between the best and worst algorithms was 0.09, 0.09, 0.07, 0.23, and 0.07 for accuracy, AUC, precision, sensitivity, and specificity, respectively (Figure 2C, Appendix A).

The coefficient magnitudes of LASSO and mboost were on average 31.8% and 42.7% smaller than its refitted magnitudes for the outcome of NP (Table 2); 33.6% and 45.9% for AP (Table 3); and 10.6% and 29.1% for disability (Table 4). The predictor that was selected by all algorithms with the largest coefficient magnitude was NRT intervention (β = from 1.987 to 2.296) for the outcome of NP (Table 2); NRT intervention (β = from 2.639 to 3.554) for the outcome of AP (Table 3); and imaging findings: spinal stenosis (β = from −1.331 to −1.763) for the outcome of disability (Table 4).

## 4. Discussion

Variable selection remains a crucial methodological tool in prognostic factor research when building statistical models [12,38]. Despite the emergence of ML algorithms in modern prediction analytics, few studies have compared newer ML algorithms with traditional stepwise regression in their difference in variable selection and influence on prediction performance. In contrast to our first hypothesis, stepwise regression using unadjusted *p* values did not result in the densest model. Also, the model with the poorest prediction performance was stepwise regression with adjusted *p* values, particularly for the outcome of disability. Qualitative inspection of the performance metrics and coefficient selection suggests that MuARS provides the optimal balance between model sparsity and high predictive performance.

The only studies that have compared different variable selection strategies in clinical predictive modelling have done so in diabetes (*n* = 803) [23], paediatric kidney injury (*n* = 6564) [39], and general hospitalised patient (*n* = 269,999) research [39]. One study reported that both forward and backward selection using *p*-value thresholds resulted in the sparsest model, compared to filter-based and wrapper-based (e.g., Stepwise AIC) selection methods [23]. This is in line with the findings of the present study, where we found that stepwise regression using *p* values, adjusted or unadjusted, resulted in a sparser model than using AIC. One study reported that gradient-boosted variable selection resulted in the sparsest model when compared to stepwise regression using *p*-value [39]. However, the gradient-boosted variable selection algorithm used a forest model with 500 trees, making it difficult to assess the univariate effects of the predictors [39]. No comparison was performed against other embedded methods like in the present study [23].

Some predictors were selected by the six algorithms that were not identified by either of the two *p*-value based methods, which was supported by a previous review reporting that a significance level of only 0.05 used in stepwise regression could miss important prognostic factors in the model [10]. For example, the predictor of “Time since first episode (years)—10 years” was not selected using stepPAdj for the outcome of NP, but a longer duration of complaints at baseline has been reported to have strong evidence as a prognostic factor for persistent pain [7]. Also, baseline disability was identified by six algorithms other than either of the two *p*-value based methods, and this was supported by a review that found strong evidence for the role of baseline functional limitations as a prognostic factor for persistent disability [7]. A disadvantage of a sparse model is not only that important prognostic information may be lost, but the predictive performance of the model also suffers, like the stepPAdj for the outcome of disability.

Our findings that the number of variables selected was closely similar between LASSO and BestSubset was supported by a previous study [40]. Another study reported that AIC selection mimics *p*-value selection, but with a significance level of roughly 0.15 (instead of 0.05), and so is more conservative with removing variables [41], which we found in the present study. Both MCP and LASSO try to approximate the best subset selection [40], whilst mboost is also a form of LASSO if the step size (learning rate) goes to zero (becomes very small) [42]. MuARS in turn performs a very similar procedure to mboost (but also with a backward step) [20]. The added backward step in MuARS could result in more variables removed, compared to mboost. 

Uncertainty in any variable selection method is selection stability [43]. Combining bootstrap resampling or subsampling with any statistical or ML algorithm has been used in past research to determine the frequency of selection of different variables on a different random subset of the original sample [8]. In the present study, we propose another method of quantifying selection stability by determining the frequency of variables selected using different algorithms [44]. In the original study, the predictors of “having undergone a NRT intervention”, “chronicity”, “baseline arm pain”, and “employment status” had a frequency of ≥90% of being selected across 100 bootstrapped samples [8]. These highly stable predictors were similarly selected with a high frequency across the investigated algorithms, which suggests that highly important variables will get selected more frequently across different samples and algorithms. Future studies investigating ensemble methods to combine multiple strategies to understand variable selection stability will be essential as a means of building prediction models that balance predictive performance and sparsity.

The present study did not investigate all possible types of ML algorithms with embedded variable selection capacity. A notable exclusion is classification or regression trees [45,46]. Although tree-based models are interpretable, the present study focuses on parametrically based algorithms to provide a comparison of not only the selection of the variable but also the magnitude and sign of the beta coefficients. A potential disadvantage of tree-based models in prognostic models is the poorer generalisability to new data, i.e., high variance compared to other algorithms [45,46]. Both mboost and MuARS can model nonlinear relationships and include interactions between variables during the model fitting process, which may further optimise the balance between predictive performance and sparsity in prognostic modelling. In the present study, we did not provide statistical inference results (e.g., standard error, confidence intervals) on the selected variables [47,48]. Valid post-selection inference is challenging and still a very active area of research, given that the use of data-driven methods introduces additional uncertainty, which invalidates classical inference techniques [47,48].

## 5. Conclusions

Different statistical and ML algorithms produced similar prediction performances but resulted in a different number of variables selected. Traditional stepwise regression based on *p*-values could miss selecting variables selected by all other ML algorithms. The MuARS appears to provide a good balance between model sparsity whilst retaining high predictive performance across outcomes. Algorithms like MuARS and mboost can model (non)linear relationships, as well as predictor interactions, which could better estimate the relationship between prognostic factors and clinical outcomes. Rather than relying on any single algorithm, confidence in the variable selection may be increased by using multiple algorithms.

## Figures and Tables

**Figure 1 jcm-12-06232-f001:**
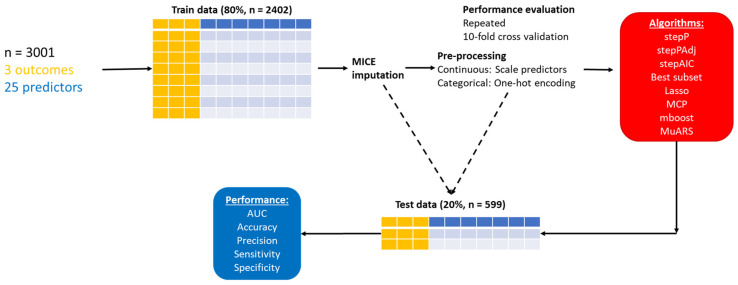
Overview of workflow. Abbreviations: stepP: stepwise logistic regression based on *p* values with no adjustment; stepPAdj: stepwise logistic regression based on *p* values with adjustment; stepAIC: stepwise logistic regression based on AIC; Best subset: best subset regression; LASSO: least absolute shrinkage and selection operator; MuARS: multivariate adaptive regression spline; MCP: Minimax concave penalty; mboost: model-based boosting; area under the receiver operating characteristic curve (AUC).

**Figure 2 jcm-12-06232-f002:**
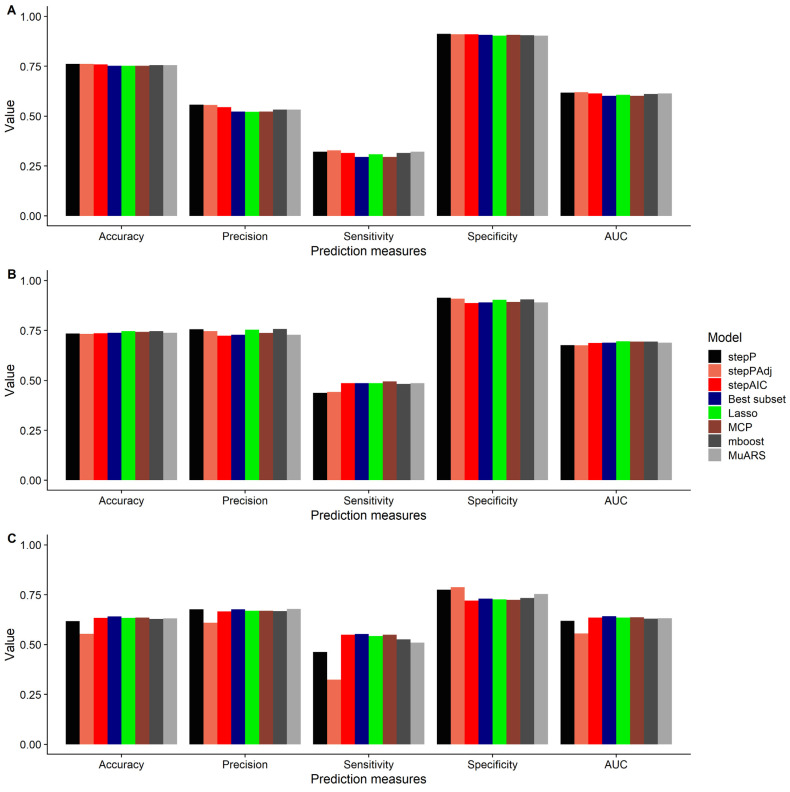
Predictive performance of eight algorithms for the clinical outcomes of (**A**) neck pain improvement, (**B**) arm pain improvement, and (**C**) disability improvement. Abbreviations. stepP: stepwise logistic regression based on *p* values with no adjustment; stepPAdj: stepwise logistic regression based on *p* values with adjustment; stepAIC: stepwise logistic regression based on AIC; BestSubset: best subset regression; LASSO: least absolute shrinkage and selection operator; MuARS: multivariate adaptive regression spline; MCP: Minimax concave penalty; mboost: model-based boosting; area under the receiver operating characteristic curve (AUC).

**Table 1 jcm-12-06232-t001:** Descriptive characteristics of participants (*n* = 3001). Continuous variables are summarised as mean (one standard deviation). Categorical variables are summarised as count (% frequency).

Variable	Total
Neck pain improvement	
N-Miss	238
No	757 (27.4)
Yes	2006 (72.6)
Arm pain improvement	
N-Miss	1061
No	568 (29.28)
Yes	1372 (70.72)
Disability improvement	
N-Miss	1796
No	600 (49.79)
Yes	605 (50.21)
Sex	
N-Miss	48
Male	726 (24.59)
Female	2227 (75.41)
Age (years)	
N-Miss	21
Mean (SD)	50.29 (15.86)
Employment	
N-Miss	376
Not applicable	1199 (45.68)
Not working	197 (7.5)
Working	1229 (46.82)
Pain duration (days)	
N-Miss	165
Mean (SD)	493.4 (989.43)
Time since first episode (years)	
N-Miss	120
<1	648 (22.49)
1–5	984 (34.15)
5–10	677 (23.5)
>10	572 (19.85)
Chronicity	
Acute	971 (32.36)
Chronic	2030 (67.64)
Baseline neck pain	
N-Miss	28
Mean (SD)	6.56 (2.25)
Baseline arm pain	
N-Miss	80
Mean (SD)	4.47 (3.38)
Baseline disability	
N-Miss	1194
Mean (SD)	30.84 (22.41)
Xray diagnosis	
No	2302 (76.71)
Yes	699 (23.29)
MRI diagnosis	
No	2399 (79.94)
Yes	602 (20.06)
Imaging findings of disc degeneration	
No	1666 (55.51)
Yes	1335 (44.49)
Imaging findings of facet degeneration	
No	2771 (92.34)
Yes	230 (7.66)
Imaging findings of scoliosis	
No	2866 (95.5)
Yes	135 (4.5)
Imaging findings of spinal stenosis	
No	2938 (97.9)
Yes	63 (2.1)
Imaging findings of disc protrusion	
No	2731 (91)
Yes	270 (9)
Imaging findings of disc herniation	
No	2483 (82.74)
Yes	518 (17.26)
Clinical diagnosis	
Disc protrusion/herniation	665 (22.16)
Spinal stenosis	63 (2.1)
Non-specific	2273 (75.74)
Pharmacological: analgesics	
No	1042 (34.72)
Yes	1959 (65.28)
Pharmacological: NSAIDS	
No	1175 (39.15)
Yes	1826 (60.85)
Pharmacological: steroids	
No	2811 (93.67)
Yes	190 (6.33)
Pharmacological: muscle relaxants	
No	2265 (75.47)
Yes	736 (24.53)
Pharmacological: opioids	
No	2949 (98.27)
Yes	52 (1.73)
Pharmacological: other	
No	2328 (77.57)
Yes	673 (22.43)
Nonpharmacological treatment	
No	2587 (86.2)
Yes	414 (13.8)
Neuroreflexotherapy	
No	421 (14.03)
Yes	2580 (85.97)

Abbreviations: N-miss—number of missing data, SD—one standard deviation.

**Table 2 jcm-12-06232-t002:** Beta coefficients of selected variables for the outcome of neck pain.

Variables	stepP	stepPAdj	stepAIC	Best Subset	LASSO	LASSO Refit	MCP	mboost	mboost Refit	MuARS	Number
Sex—female	−0.244		−0.200	−0.198	−0.149	−0.210	−0.198	−0.128	−0.210		6
Age (years)				0.090	0.019	0.070	0.090	0.002	0.070		4
**Employment—not working**	**−0.461**	**−0.521**	**−0.538**	**−0.497**	**−0.437**	**−0.495**	**−0.498**	**−0.416**	**−0.495**	**−0.531**	**8**
Employment—working	0.150	0.127	0.163	0.210	0.141	0.180	0.210	0.125	0.180		7
Duration of pain (days)			0.084	0.084	0.030	0.071	0.084	0.017	0.071		5
Time since first episode (years)—1–5	−0.359		−0.366	−0.388	−0.144	−0.241	−0.387	−0.112	−0.241		6
Time since first episode (years)—5–10	−0.233		−0.234	−0.270			−0.269				4
Time since first episode (years)—>10	−0.569		−0.599	−0.648	−0.314	−0.469	−0.648	−0.260	−0.469	−0.312	7
Chronicity—chronic		−0.555	−0.540	−0.527	−0.411	−0.536	−0.528	−0.366	−0.536	−0.537	7
Baseline intensity of neck pain	0.163		0.236	0.225	0.178	0.222	0.225	0.161	0.222	0.240	7
Baseline intensity of arm pain			−0.165	−0.163	−0.115	−0.162	−0.163	−0.099	−0.162	−0.182	6
**Baseline disability**	**−0.247**	**−0.237**	**−0.224**	**−0.217**	**−0.201**	**−0.216**	**−0.217**	**−0.193**	**−0.216**	**−0.270**	**8**
Diagnostic procedure: X-ray—yes			0.211	0.212	0.167	0.205	0.212	0.153	0.205		5
Diagnostic procedure: MRI-yes				−0.052			−0.052				2
Imaging findings: disc degeneration—yes	−0.242	−0.293		−0.191	−0.144	−0.185	−0.191	−0.129	−0.185		6
Imaging findings: facet joint degeneration—yes			−0.449	−0.427	−0.358	−0.441	−0.426	−0.331	−0.441	−0.414	6
Imaging findings: scoliosis—yes			0.447	0.469	0.301	0.460	0.469	0.247	0.460		5
Imaging findings: spinal stenosis—yes				0.133			0.132				2
Imaging findings: disc protrusion—yes			−0.275	−0.228	−0.207	−0.234	−0.227	−0.198	−0.234		5
Imaging findings: disc herniation—yes	−0.313		−0.302	−0.253	−0.234	−0.258	−0.253	−0.223	−0.258	−0.305	7
Pharmacological treatment: analgesics—yes				0.007							1
Pharmacological treatment: NSAIDs—yes			0.161	0.146	0.082	0.137	0.149	0.063	0.137		5
Pharmacological treatment: steroids—yes				−0.207	−0.047	−0.161	−0.206	−0.012	−0.161		4
Pharmacological treatment: muscle relaxants—yes				0.136	0.054	0.127	0.137	0.029	0.127		4
Pharmacological treatment: opioids—yes				0.251	0.102	0.305	0.252	0.037	0.305		4
Pharmacological treatment: other treatments—yes				0.089			0.089				2
Nonpharmacological treatments—yes				−0.059			−0.059				2
**NRT**	**1.987**	**2.343**	**2.239**	**2.186**	**2.031**	**2.136**	**2.186**	**1.987**	**2.136**	**2.296**	**8**
Number	11	6	18	28	22	22	27	22	22	9	

Text in bold indicates variables selected by all algorithms. Abbreviations. stepP: stepwise logistic regression based on *p* values with no adjustment; stepPAdj: stepwise logistic regression based on *p* values with adjustment; stepAIC: Stepwise logistic regression based on AIC; BestSubset: best subset regression; LASSO: least absolute shrinkage and selection operator; MuARS: multivariate adaptive regression spline; MCP: Minimax concave penalty; mboost: model-based boosting; MRI: magnetic resonance imaging; NSAIDS: nonsteroidal anti-inflammatory drug: NRT: neuroreflexotherapy.

**Table 3 jcm-12-06232-t003:** Beta coefficients of selected variables for the outcome of arm pain.

Variables	stepP	stepPAdj	stepAIC	Best Subset	LASSO	LASSO Refit	MCP	mboost	mboost Refit	MuARS	Number
Sex—female											0
Age (years)											0
Employment—not working	−0.538		−0.454	−0.429	−0.364	−0.481	−0.458	−0.312	−0.486	−0.429	7
Employment—working	0.189		−0.008					0.026	−0.025		3
Duration of pain (days)					0.010	0.055	0.013				2
Time since first episode (years)—1–5			−0.261		−0.064	−0.273	−0.262	−0.003	−0.260		4
Time since first episode (years)—5–10			−0.533	−0.350	−0.314	−0.559	−0.538	−0.238	−0.539	−0.350	6
Time since first episode (years)—>10			−0.726	−0.542	−0.511	−0.762	−0.732	−0.430	−0.729	−0.542	6
Chronicity—chronic			−0.529	−0.538	−0.462	−0.572	−0.541	−0.425	−0.536	−0.538	6
**Baseline intensity of neck pain**	**−0.428**	**−0.407**	**−0.384**	**−0.384**	**−0.318**	**−0.381**	**−0.381**	**−0.296**	**−0.381**	**−0.384**	**8**
**Baseline intensity of arm pain**	**0.623**	**0.608**	**0.744**	**0.742**	**0.689**	**0.748**	**0.744**	**0.666**	**0.747**	**0.742**	**8**
Baseline disability			−0.336	−0.339	−0.334	−0.363	−0.346	−0.321	−0.360	−0.339	6
Diagnostic procedure: X-ray—yes											0
Diagnostic procedure: MRI—yes											0
Imaging findings: disc degeneration—yes			−0.307	−0.317	−0.271	−0.280	−0.300	−0.260	−0.293	−0.317	6
Imaging findings: facet joint degeneration—yes					−0.038	−0.068		−0.029	−0.071		2
Imaging findings: scoliosis—yes					0.082	0.198	0.014	0.044	0.191		3
Imaging findings: spinal stenosis—yes					−0.220	−0.304	−0.149	−0.187	−0.321		3
Imaging findings: disc protrusion—yes					0.131	0.242	0.133	0.098	0.229		3
Imaging findings: disc herniation—yes			−0.353	−0.350	−0.308	−0.358	−0.355	−0.292	−0.351	−0.350	6
Pharmacological treatment: analgesics—yes			0.329	0.321	0.191	0.234	0.288	0.177	0.229	0.321	6
Pharmacological treatment: NSAIDs—yes	0.227				0.111	0.141	0.063	0.099	0.134		4
Pharmacological treatment: steroids—yes											0
Pharmacological treatment: muscle relaxants—yes					0.059	0.104	0.039	0.042	0.108		3
Pharmacological treatment: opioids-yes			0.792	0.793	0.605	0.792	0.731	0.547	0.796	0.793	6
Pharmacological treatment: other treatments—yes	−0.262	−0.310									2
Nonpharmacological treatments—yes					0.008	0.053					1
**NRT**	**2.639**	**2.695**	**3.525**	**3.447**	**3.218**	**3.549**	**3.534**	**3.101**	**3.554**	**3.447**	**8**
Number	7	4	14	12	21	21	19	20	20	12	

Text in bold indicates variables selected by all algorithms. Abbreviations. stepP: stepwise logistic regression based on *p* values with no adjustment; stepPAdj: stepwise logistic regression based on *p* values with adjustment; stepAIC: stepwise logistic regression based on AIC; BestSubset: best subset regression; LASSO: least absolute shrinkage and selection operator; MuARS: multivariate adaptive regression spline; MCP: Minimax concave penalty; mboost: model-based boosting; MRI: magnetic resonance imaging; NSAIDS: nonsteroidal anti-inflammatory drug: NRT: neuroreflexotherapy.

**Table 4 jcm-12-06232-t004:** Beta coefficients of selected variables for the outcome of disability.

Variables	stepP	stepPAdj	stepAIC	Best Subset	LASSO	LASSO Refit	MCP	mboost	mboost Refit	MuARS	Number
Sex—female	0.232			0.108	0.096	0.108	0.099	0.063	0.108		5
**Age (years)**	**0.193**	**0.159**	**0.198**	**0.204**	**0.186**	**0.203**	**0.201**	**0.137**	**0.203**	**0.157**	**8**
Employment—not working	0.149	0.042	−0.327	−0.312	−0.291	−0.310	−0.310	−0.236	−0.309		7
**Employment—working**	**0.422**	**0.397**	**0.276**	**0.276**	**0.264**	**0.278**	**0.278**	**0.223**	**0.278**	**0.252**	**8**
Duration of pain (days)	−0.151		−0.135	−0.139	−0.139	−0.142	−0.141	−0.129	−0.142	−0.158	7
Time since first episode (years)—1–5			−0.431	−0.440	−0.394	−0.438	−0.438	−0.265	−0.438		5
Time since first episode (years)—5–10			−0.385	−0.395	−0.341	−0.393	−0.393	−0.188	−0.393		5
Time since first episode (years)—>10			−0.474	−0.482	−0.421	−0.479	−0.477	−0.251	−0.479		5
Chronicity—chronic			−0.389	−0.400	−0.386	−0.400	−0.397	−0.345	−0.400	−0.405	6
Baseline intensity of neck pain			0.096	0.090	0.084	0.089	0.088	0.068	0.089		5
Baseline intensity of arm pain	−0.175		−0.386	−0.394	−0.381	−0.393	−0.394	−0.344	−0.393	−0.359	7
Baseline disability			0.426	0.433	0.421	0.433	0.432	0.387	0.433	0.447	6
Diagnostic procedure: X-ray—yes	0.357		0.305	0.296	0.289	0.299	0.298	0.257	0.300	0.294	7
Diagnostic procedure: MRI—yes	0.270				0.000	0.011					2
Imaging findings: disc degeneration—yes			−0.338	−0.319	−0.303	−0.318	−0.322	−0.256	−0.319	−0.296	6
**Imaging findings: facet joint degeneration—yes**	**−0.820**	**−0.756**	**−0.770**	**−0.790**	**−0.766**	**−0.790**	**−0.786**	**−0.699**	**−0.790**	**−0.776**	**8**
**Imaging findings: scoliosis—yes**	**0.588**	**0.653**	**0.547**	**0.543**	**0.509**	**0.540**	**0.538**	**0.417**	**0.540**	**0.493**	**8**
**Imaging findings: spinal stenosis—yes**	**−1.420**	**−1.331**	**−1.777**	**−1.761**	**−1.703**	**−1.763**	**−1.758**	**−1.540**	**−1.761**	**−1.628**	**8**
**Imaging findings: disc protrusion—yes**	**−0.640**	**−0.654**	**−0.676**	**−0.679**	**−0.669**	**−0.683**	**−0.684**	**−0.631**	**−0.682**	**−0.692**	**8**
Imaging findings: disc herniation—yes			0.211	0.211	0.187	0.209	0.212	0.114	0.212		5
Pharmacological treatment: analgesics—yes				−0.075	−0.030	−0.039		−0.001	−0.039		3
Pharmacological treatment: NSAIDs—yes					−0.060	−0.069	−0.076	−0.037	−0.068		3
Pharmacological treatment: steroids—yes				0.297	0.271	0.296	0.293	0.198	0.296	0.419	5
Pharmacological treatment: muscle relaxants—yes		0.373	0.227	0.227	0.225	0.239	0.235	0.180	0.239		6
Pharmacological treatment: opioids—yes				−0.226	−0.190	−0.224	−0.134	−0.089	−0.224		4
Pharmacological treatment: other treatments—yes			0.262	0.198	0.193	0.201	0.198	0.166	0.203		5
Nonpharmacological treatments—yes			−0.203	−0.225	−0.200	−0.222	−0.223	−0.141	−0.220		5
NRT			1.200	1.254	1.224	1.252	1.253	1.141	1.253	1.238	6
Number	12	8	22	26	28	28	26	27	27	14	

Text in bold indicates variables selected by all algorithms. Abbreviations. stepP: stepwise logistic regression based on *p* values with no adjustment; stepPAdj: stepwise logistic regression based on *p* values with adjustment; stepAIC: stepwise logistic regression based on AIC; BestSubset: best subset regression; LASSO: least absolute shrinkage and selection operator; MuARS: multivariate adaptive regression spline; MCP: Minimax concave penalty; mboost: model-based boosting; MRI: magnetic resonance imaging; NSAIDS: nonsteroidal anti-inflammatory drug: NRT: neuroreflexotherapy.

## Data Availability

The datasets analysed during the current study are available from the author (F.M.K) on reasonable request. The codes used for the present study are included in the lead author’s public repository (https://github.com/bernard-liew/spanish_data_repo accessed on 18 September 2023).

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
