# Peer review of "Automatic Variable Selection Algorithms in Prognostic Factor Research in Neck Pain"

_jcm, 2023, doi:10.3390/jcm12196232_

Round 1
Reviewer 1 Report
Thank you for the invitation to review this article and congratulations to the authors.
I encourage the authors to reduce the copy rate from line 81 to 142, as I find it a very interesting article.
of articles such as:
Liew, B.X.W., Kovacs, F.M., Rügamer, D. et al. Machine learning versus logistic regression for prognostic modelling in individuals with non-specific neck pain. Eur Spine J 31, 2082-2091 (2022). https://doi.org/10.1007/s00586-022-07188-w
Author Response
Dear Reviewer, please see the uploaded Response attachment for my responses. Thank you for taking the time to review.

Reviewer 2 Report
Hello,
Thank you for allowing me the opportunity to review this paper. Your research really interested me, as my core field across multiple disciplines is in statistics. Machine learning and AI are both on the rise throughout multiple industries, there is no question about this.
I believe your research provides a really important view - we hear a lot about machine learning nowadays, but how is this different to the traditional regression approaches, and also is it really worth going down the machine learning path? Your paper answers these questions in this particular field. I have a background in R as well and have taken the time to review your code - it is so refreshing to see such transparency. Thank you; I felt it was worth reviewing.
Your paper is well considered, with all the details I’m interested in. All the expected sections and associated detail within each. Interesting conclusion you come to on using multivariate regression splines. In my experience I’d be tempted to agree with you that the use of splines do offer a performance uplift, without adding too much complexity and generally you are still able to explain the results themselves. Interesting with the beta coefficient variation between models, but reassuring that no matter the model you are looking at, the direction of travel is the same across and remains intuitive. This is a useful finding in itself.
All I would say is that, and I’m sure you are already aware of this, but this kind of research whilst I understand the clinical impact and importance of course, I know that also statistical researchers would be very interested in what you’ve produced.
I’m excited to see further work from you.
Author Response

(The authors gave the same response as above.)

Reviewer 3 Report
The authors have addressed an interesting and important problem in the field of bioinformatics and machine learning. They have compared different variable selection algorithms in machine learning and statistics. It provide the researchers an overview of each technique and help to choose the appropriate technique according their problem statement. I have following comments:
1) It is always better to provide the full forms rather than abbreviations in the abstract, which can help the readers to understand the paper better. Please provide the full form NRT in the abstract.
2) Line 16 "Twenty-five variables (28 parameters) were included as 16 predictors." is confusing if the authors has 25 variables or 28. Please clarify this statement in the abstract.
3) It would be very informative if authors can provide the rationale behind choosing the algorithms for comparison in this study.
4) Please mention the full forms of the abbreviations used in table 1 at its footer, such as N-Miss, SD, etc.
5) On observing Figure 1, 2, and 3, it seems that all the algorithms are working equivalent. How does authors decided the best algorithm. Is number of predictors only criteria should be used for the decision.
6) The conclusion is weak and not able to provide the overall idea of the study. I would suggest authors to rewrite a strong conclusion for this study.
7) The quality of the figures are not adequate. Please provide the high resolution images.
8) I would suggest authors to provide an image which exhibits the overall design or workflow of the study. Which can capture the overall idea of the study.
9) GitHub or GitLab is a widely accepted platforms for code sharing. I would suggest authors to provide their codes and data on them.
10) Very minute differences in the performance of each algorithm are not captured by the figures provided by the authors. If possible I would be great to provide the performance measures in the tables so that readers can appreciate the differences in the performances.
Author Response

(The authors gave the same response as above.)

Reviewer 4 Report
The authors have conducted a relatively comprehensive examination of several variable selection strategies, focusing on the application of predictive modeling of neck pain recovery. That said, the study as currently presented has several shortcomings that must be addressed prior to publication. I have the following requested revisions:
1. The authors state two hypotheses: (1) that traditional stepwise regression using unadjusted P values would lead to the least sparse model, and (2) that the prediction performance of traditional stepwise regression using unadjusted P values would be the poorest compared to the remaining ML algorithms assessed. In the revised manuscript, a rationale and justification for these hypotheses should be included.
2. Bayesian approaches are continuing to gain prominence in model/variable selection studies, but such approaches are entirely neglected in the present study. For completeness, model/variable selection results based on Bayes' factor and the closely related Bayesian information criterion should be included, or otherwise their absence should be justified.
3. In Figures 1-3 (or at least as supplementary material), it would be clearer if the different prediction measures were shown on different subplots, each with their own vertical axis limits to emphasize differences in performance between the models. As currently shown, it is difficult to assess differences in performance, since all models appear to be approximately equal within each prediction measure.
4. On line 158, the abbreviation AUC should be defined upon its first use.
5. In the captions of Figures 1-3, "receiver operating curve" should be corrected to "receiver operating characteristic (ROC) curve."
6. Since AUC is considered as a prediction measure, a figure showing the ROC curves for the different models should be included, preferably in the main manuscript or otherwise as supplementary material.
7. As supplementary material, uncertainty quantification results should be presented for all included variables in each of the models considered. These should include both confidence intervals for individual variables as well as estimated parameter covariance or correlation matrices.
Author Response

(The authors gave the same response as above.)

Round 2
Reviewer 1 Report
Thank you for your efforts
Reviewer 3 Report
The authors have addressed my comments satisfactorily.
Reviewer 4 Report
All previous comments and requests for revisions have been adequately addressed.